# *GsMAS1* Encoding a MADS-box Transcription Factor Enhances the Tolerance to Aluminum Stress in *Arabidopsis thaliana*

**DOI:** 10.3390/ijms21062004

**Published:** 2020-03-15

**Authors:** Xiao Zhang, Lu Li, Ce Yang, Yanbo Cheng, Zhenzhen Han, Zhandong Cai, Hai Nian, Qibin Ma

**Affiliations:** 1The State Key Laboratory for Conservation and Utilization of Subtropical Agro-bioresources, South China Agricultural University, Guangzhou 510642, China; XZHANG916@163.com (X.Z.); 2019201044@njau.edu.cn (L.L.); yangce26@outlook.com (C.Y.); ybcheng@scau.edu.cn (Y.C.); zhenzhenH1@outlook.com (Z.H.); zdcai@stu.scau.edu.cn (Z.C.); 2The Guangdong Provincial Laboratory of Lingnan Modern Agricultural Science and Technology, South China Agricultural University, Guangzhou 510642, China; 3The Key Laboratory of Plant Molecular Breeding of Guangdong Province, College of Agriculture, South China Agricultural University, Guangzhou 510642, China; 4The Guangdong Subcenter of National Center for Soybean Improvement, College of Agriculture, South China Agricultural University, Guangzhou 510642, China; 5The National Engineering Research Center of Plant Space Breeding, South China Agricultural University, Guangzhou 510642, China

**Keywords:** GsMAS1, MADS, Al stress, Glycine Soja, Arabidopsis thaliana

## Abstract

The MADS-box transcription factors (TFs) are essential in regulating plant growth and development, and conferring abiotic and metal stress resistance. This study aims to investigate *GsMAS1* function in conferring tolerance to aluminum stress in Arabidopsis. The *GsMAS1* from the wild soybean BW69 line encodes a MADS-box transcription factor in *Glycine soja* by bioinformatics analysis. The putative GsMAS1 protein was localized in the nucleus. The *GsMAS1* gene was rich in soybean roots presenting a constitutive expression pattern and induced by aluminum stress with a concentration-time specific pattern. The analysis of phenotypic observation demonstrated that overexpression of *GsMAS1* enhanced the tolerance of Arabidopsis plants to aluminum (Al) stress with larger values of relative root length and higher proline accumulation compared to those of wild type at the AlCl_3_ treatments. The genes and/or pathways regulated by *GsMAS1* were further investigated under Al stress by qRT-PCR. The results indicated that six genes resistant to Al stress were upregulated, whereas *AtALMT1* and *STOP2* were significantly activated by Al stress and *GsMAS1* overexpression. After treatment of 50 μM AlCl_3_, the RNA abundance of *AtALMT1* and *STOP2* went up to 17-fold and 37-fold than those in wild type, respectively. Whereas the RNA transcripts of *AtALMT1* and *STOP2* were much higher than those in wild type with over 82% and 67% of relative expression in *GsMAS1* transgenic plants, respectively. In short, the results suggest that *GsMAS1* may increase resistance to Al toxicity through certain pathways related to Al stress in Arabidopsis.

## 1. Introduction

The metal element most commonly found in soil is aluminum (Al) with an average content of about 7.45% [1]. Aluminum has a toxic effect on plants by limiting the growth of roots depending on its existing form of Al^3+^ ion in acidic soil at pH ≤ 5 [2,3]. The Al^3+^ ion at low concentrations (micromole levels) can quickly inhibit root growth resulting in reduced crop yield [4]. Therefore, it is believed that aluminum in acidic soil is the main factor limiting crop growth [5,6,7,8]; however, it is also a key factor limiting the growth and development of soybean [9].

The root tip is a sensitive part of the plant that responds to Al stress, and the initial manifestation of Al toxicity is that the elongation of the root tip is inhibited [8]. Previous reports found that the accumulation of Al^3+^ ion was mainly in 0–5 mm root tips with the richest section from the root tips of 2–3 mm length in crops such as wheat, maize, and peas [10,11,12,13]. Therefore, the root elongation and root Al^3+^ content in plants are useful in measuring plant aluminum toxicity [14].

The mechanisms of plant resistance to aluminum toxicity are external rejection and internal detoxification [15]. The mechanism of external rejection mainly takes place in the plant roots to prevent aluminum ions entering the root tip cells, whereas the detoxification mechanism has four main components: organic acid (OA) chelate aluminum ions, induced pH barrier, cell wall of aluminum fixation, and Al^3+^ transmembrane outflow. The aluminum toxicity caused by aluminum ions that have entered the plant cells can be alleviated mainly through the chelation of organic acids, the aluminum compartment in the vacuole and the combination of protein and other forms of internal detoxification [16,17,18]. Organic acid transporters including aluminum-activated malate transporter1 (ALMT1) and other members of the multidrug and toxic compound extrusion (MATE) citrate transporter family are reported to be involved in the external rejection of the aluminum toxicity mechanism in plants [19,20,21,22]. A recent study suggested that transcription factors also play certain roles in Al stress during the process of plant growth and development, for example, *WRKY46* as a transcriptional repressor of *ALMT1* regulated aluminum-induced malate secretion in Arabidopsis [23]; *WRKY22* improved tolerance to Al stress through activation of *FRDL4* and enhancement of citrate secretion in rice [24]. *STOP1* encoding a Cys2/His2 type zinc-finger transcription factor played critical roles in regulating ALMT1 and other genes to protect Arabidopsis from proton and aluminum toxicity [25,26,27,28], whereas several genes regulated by STOP1 can be activated by STOP2 under aluminum stress which is a STOP1 homolog in Arabidopsis [29]. In the meantime, homologous genes of *STOP1*, widely found in other plants such as *NtSTOP1* in tobacco, *GmSTOP1* in soybean and *SbSTOP1*-like genes in sweet sorghum, endow plants resistance to aluminum toxicity [30,31,32]. In addition, Agamous-like MADS-box protein AGL62 and NAC domain-containing protein 100 are the leading transcription factors (TFs) contributing flax tolerance to Al stress through regulation of plant growth and development [33].

The MADS family found in fungi, animals and plants possesses a highly conserved DNA-binding domain named MADS at N-terminal [34,35,36]. Based on the phylogenetic analysis, the MADS gene family created through a gene duplication occurred before the divergence of plants and animals can be divided into two large lineages of type I and type II [37,38]. In plants, the typical difference of MADS genes between type II and type I is that the plant MADS genes of type II have a k-domain [37,39]. The plant-special type II MADS has typical structures with MADS, I, K, and C domains, and is also known as MIKC MADS [40,41]. The MADS region contains a highly conserved element with approximately 60 amino acids and the MADS domain binds specifically to DNA [42,43]. The I region consists of between 30 and 35 amino acids containing more hydrophilic residues, and has the function of DNA specific binding [44]. The K region consists of 65–70 amino acids mediating the interaction between proteins [45]. The C region comprises about 30 hydrophobic amino acids, and the sequence and length of the lowest conservatism is to promote the formation of protein complexes, DNA binding and transcriptional activation [46,47,48].

The MADS family is an ancient and broadly studied transcription factor and plays essential roles in almost all developmental processes in plants, such as floral organ development [49,50], controlling the flowering time of plants [51], ovary development, seed-coat development [52,53], embryo development [54], determining the meristem [55], root growth [56], plant vegetative growth [57] and symbiosis induction [58], fruit development and ripening [59], silique architecture [60], modulating plant architecture and abscisic acid [61], and orchid reproductive development [62]. However, less attention is paid to stresses regulated by MADS-box genes in plants. Recent studies have shown that the MADS-box genes regulate plant resistance to abiotic stresses [63]. In tomato, flower abnormalities induced by low temperatures are associated with expression changes of MADS-box genes including TM4, TM5, TM6 and TAG1 [64]. The *OsMADS*26 overexpression can reduce the stress response to the external environment of transgenic rice and Arabidopsis [65]. Together with several other genes, *OsMADS87* could be a potential target for improving the thermal resilience of rice during reproductive development [66]. Moreover, *CaMADS* upregulated by abiotic stress in pepper functions as a positive stress-responsive transcription factor in the signaling pathways of cold, salt, and osmotic stress [67]. Genome-wide identification indicated that there were many MADS-box genes responding to abiotic stresses such as salt, drought, heat, and cold stress with differential expression levels in rice [68,69], *Brachypodium distachyon* seedlings [70], cabbage seedlings [71], *Brassica rapa* [70], and bread wheat [72]. Therefore, the recent reports suggested that MADS-box genes are key components of gene regulatory networks involved in responding to stress changes and developmental plasticity in plants [73].

In soybean, the MADS-box family with 163 putative members is also one of the biggest transcription factor superfamilies that regulates a variety of biological functions [38,72,74]. In recent decades, only several members of soybean MADS-box genes have been investigated such as *GmNMH7*, *GmMADS29*, *SOC1*, *GmGAL1*, *GmGAL2*, *GmAGL15*, *GmMADS28,* and *GmSEP1* [75,76,77,78,79]. Among them, *GmNMH7* is a photoperiod-dependent gene involved in flower development in soybean flowering reversion system [75]. *GmAGL1* plays important roles in both floral organ identity and fruit dehiscence, and promotes flowering through the photoperiod pathway [80,81]. A gain-of function MADS-box gene, *Dt2*, determines stem growth by repressing *Dt1* expression to promote early conversion of the shoot apical meristems into reproductive inflorescences [82]. Furthermore, *GmAGL15* can enhance the development of somatic embryogenesis both in Arabidopsis and soybean by promoting a dedifferentiated state, and partially by the control of ethylene biosynthesis and response [76,83,84,85]. Heterologous expression of *GmMADS28* in tobacco resulted in abnormal flower organ development in transgenic tobacco plants and promoted the flowering and dwarfing of transgenic plants in advance [77]. Mainly involved in the development of seeds in soybean, *GmSEP1* is related to the characteristics of floral organs [78]. Heterologous expression of *GmGAL2* accelerate flowering in Arabidopsis [79]. Other MADS-box genes such as *GmFLUa*, *GmAGL6*, and *GmSOC1* also play essential roles in reproductive transition, flowering, the floral initiation process and leaf development in soybean [86,87,88]. However, there are few studies on the MADS-box genes related to abiotic stresses and/or heavy metal stress in soybean. In this study, the *GsMAS1* encoding a MADS-box transcription factor was cloned from the wild soybean BW69 line. *GsMAS1* showing a constitutive expression pattern was accumulated in roots and induced by Al stress. Heterologous transformation of *GsMAS1* in Arabidopsis was used to investigate the molecular mechanism of the *GsMAS1* gene responding to aluminum stress.

## 2. Results

### 2.1. Bioinformatics Analysis of GsMAS1 Gene

Based on data of the gene expression profile of the Al-resistant BW69 line of *Glycine soja* [89], an aluminum-induced gene from the gene expression profile was screened from the National Center for Biotechnology Information (NCBI) database using the sequence information. The sequence alignment showed that the gene had exactly the same sequence of open reading frame (ORF) as *GmNMHC5* (accession: NM_001254560.3) [90], which is located on soybean chromosome 13. The candidate gene was cloned from the BW69 line of *Glycine soja* by specific primers using the cDNA sequence of *GmNMHC5* as a reference. The candidate gene confirmed by the sequence results was then designated as *GsMAS1* (which encodes a MADS-box transcription factor tolerant to aluminum stress 1 in *Glycine soja*). The full cDNA sequence of *GsMAS1* from RT-PCR was 1000 bp with an ORF of 726 bp which encodes 241 amino acids with MW of 27.768 kDa (Additional file 1, Appendix A). The *GsMAS1* gene was induced by Al stress (25 μM AlCl_3_, 0.5 mM CaCl_2_, pH 4.3) with over 2-fold expression level from the data of the gene expression profile [89,91]. The Basic Local Alignment Search Tool (BLAST) analysis of NCBI revealed that putative GsMAS1 protein has the structural characteristics of the MADS-box transcription factor with two conservative domains (Additional file 2, Appendix A). One of them is the MADS-box domain located at the position from the 2^nd^ to the 77^th^ amino acids; the other one is the k-box domain located at the 81^st^ to the 169^th^ amino acids. The structural characteristics showed that GsMAS1 protein belongs to myocyte enhancer factor 2 (MEF2)-like/Type II subfamily of MADS (MCM1, Agamous, Deficiens), and serum response factor (SRF) box family of eukaryotic transcriptional factors (Additional file 2, Appendix A). Multiple sequence alignments of MADS-box proteins (Type II subfamily) indicated that GsMAS1 protein is highly homologous with 59.27% identity compared to those of AtAGL21, BRADI_5g12437v3, GmAGL15, GsMADS23L1 and MTR_2g009890 proteins, whereas it is less homologous (45.52% identity) with those of AtAGL19, Dt2, OsMADS8, and OsMADS20 proteins (Figure 1a). The phylogenetic analysis also indicated differences among the members of the Type II subfamily of MADS-box transcription factors in plants. As shown in Figure 1b, 35 MADS-box TF proteins from soybean, Arabidopsis, rice, short-handled grass and alfalfa play diverse functions in regulation of male and female gametophyte morphogenesis (AtAGL13), floral activator (AtAGL19), flowering (GmAGL1, GmAP1, GmFULa, GmNMH7, OsMADS8), determinate floral meristems (AtAP1, AtSEP2); semi-determinate stem growth (GmSOC1), somatic embryogenesis (GmAGL15), cell divisions (AtSHP1, AtSHP2); age-related resistance (AtSVP), root development (AtAGL21, AtAGL44, GmNMHC5) (Figure 1b). The bioinformatics analysis indicated that GsMAS1 protein has a closer affinity with some MADS-box TF proteins, such as GsNMHC5, AtAGL21, and AtAGL44 proteins. The results suggest that GsMAS1 protein may function as a MADS transcription factor in root development and/or root traits in soybean.

### 2.2. Expression Patterns of GsMAS1 in Tissues and under Aluminum Stress

To detect the tissue expression pattern of *GsMAS1*, the samples of young root, leaf, and stem from the seedlings, and flower and pod at R_2_ stage, were taken from the BW69 line of *Glycine soja*. The qRT-PCR was carried out to determine the expression levels of the *GsMAS1* gene in different tissues and organs. The qRT-PCR result showed that the *GsMAS1* gene presented a constitutive expression pattern and was abundant in roots, which was over twice of relative expression levels higher than those in other tissues and organs (Figure 2a).

To investigate the expression patterns of the *GsMAS1* response to acidic aluminum stresses, the qRT-PCR was used to analyze the transcript abundance of *GsMAS1* under AlCl_3_ treatments designed with concentration gradients and time gradients. As shown in Figure 2b, *GsMAS1* was upregulated during the process of AlCl_3_ treatments (0.5 mM CaCl_2_, pH 4.3). With the increase of AlCl_3_ concentration, the transcription abundance of *GsMAS1* increased gradually reaching the highest level at over three times that of the control at 50 μM AlCl_3_, and then declined to a lower level under high concentrations of AlCl_3_ (Figure 2b). Under different time-gradient conditions, the transcripts of *GsMAS1*was accumulated under AlCl_3_ treatment (Figure 2c). In the absence of AlCl_3_ (pH 4.3), the expression levels of *GsMAS1* at 6 h and 12 h treatments showed no significant difference compared with those of the control (pH 5.8). However, the transcription abundance of *GsMAS1* increased first and then decreased along with the treatment time of 50 μM AlCl_3_ (pH 4.3). The transcripts of *GsMAS1* reached the highest level at 6 h treatment, 3.5 times of that of the control with no significant differences at other time points of AlCl_3_ treatment (Figure 2c). The results suggest that *GsMAS1* might play a potential role in Al stress.

### 2.3. Subcellular Localization Analysis of GsMAS1 Protein

To analyze the potential function of GsMAS1 protein, subcellular localization analysis of GsMAS1 protein was carried out in Arabidopsis protoplasts. As shown in Figure 3, the eGFP (enhanced green fluorescent protein) fluorescence signal in the protoplasts transformed with pYL322-d1-eGFP plasmid was clearly distributed throughout the cells, whereas the eGFP fluorescence signal in the protoplasts transformed with YL322-d1-eGFP-GsMAS1 plasmid was only detected in the nucleus (Figure 3). The results demonstrated that GsMAS1 protein was located in the nucleus.

The vectors of pYL322-d1-eGFP and pYL322-d1-eGFP-GsMAS1 (eGFP: enhanced green fluorescent protein) were separately transformed into Arabidopsis protoplast cells using the heat-shock method. The vector-transformed cells cultured for 24 h were observed under a confocal scanning microscope (Leica, Germany) [50].

### 2.4. Molecular Identification of GsMAS1 Transgenic Arabidopsis Lines

Thirteen *GsMAS1* transgenic lines in T_3_ generation were underwent molecular identification to obtain homozygous lines. The PCR products of a specific band of 455 bp was detected in different *GsMAS1* transgenic lines (Figure 4a). The qRT-PCR was performed to determine the transcription levels of *GsMAS1* in wild type and transgenic lines using specific primers (Additional file 3, Appendix A). Six homozygous lines of *GsMAS1* overexpression were further confirmed by qRT-PCR. The results indicated that the *GsMAS1* gene was overexpressed in transgenic lines with higher levels specifically in the lines of L4, L9, and L13. No RNA transcript of *GsMAS1* was detected in wild type of Arabidopsis (Figure 4b).

### 2.5. Phenotypic Analysis of GsMAS1 Transgenic Lines

The three lines L4, L9, and L13, with a relatively high expression level of *GsMAS1*, were selected to investigate the acidic-resistant phenotypes of *GsMAS1* transgenic plants. The results showed the root elongations of Arabidopsis were inhibited under Al stress. With the increase of AlCl_3_ concentration, the growth and development of roots were inhibited to a greater extent (Figure 5). The observation results indicated that there was no difference in elongation of taproots between *GsMAS1* transgenic lines and wild-type without AlCl_3_ treatment (Figure 5a). The growth of the main roots of wild-type Arabidopsis was inhibited when treated with 50 μM AlCl_3_ (pH 4.5), whereas the main roots of transgenic Arabidopsis plants were significantly longer than those of wild type. The inhibition degree of principal roots increased significantly under the individual AlCl_3_ treatments of 100 μM, 150 μM, and 200 μM (Figure 5a,b). The relative root length (RRL) of the main root of *GsMAS1* transgenic lines was more than 70%, at 50 μM AlCl_3_, significantly higher than that of wild type (50%). Under the treatments of 100 μM and 150 μM AlCl_3_, the main root elongation of Arabidopsis was inhibited with a large decrease of RRLs both in wild type and transgenic plants. Compared to those of wild type, the roots of *GsMAS1* transgenic lines was less affected by Al stress with RRLs of over 50% and 40%, respectively (Figure 5a,b). At 200 μM AlCl_3_ treatment, there was no difference in the elongation of the main roots between *GsMAS1* transgenic lines and wild type with RRLs at about 20% (Figure 5a,b). The results of phenotypic identification indicated that overexpression of the *GsMAS1* gene enhanced the tolerance of transgenic Arabidopsis to Al stress.

### 2.6. Determination of Free Proline Content

Previous reports showed that proline accumulation can enhance maize tolerance to aluminum stress by minimizing accumulation of lipid peroxides [93]. In present study, in the absence of AlCl_3_ treatment, the content of free proline was about 145 μg·g^−1^ in both wild-type and *GsMAS1* transgenic plants with almost no difference (Figure 5c). Compared to the control, the content of free proline in Arabidopsis plants increased significantly at the treatment of 50 μM AlCl_3_. The content of free proline in wild type and *GsMAS1* transgenic Arabidopsis plants was about 230 μg·g^−1^ and 440 μg·g^−1^, respectively (Figure 5c). The determination result indicated that free proline was substantially accumulated in *GsMAS1* transgenic Arabidopsis under Al stress.

### 2.7. Expression Patterns of Al Stress Responsive Genes Regulated by GsMAS1

To make an investigation into the pathways regulated by *GsMAS1* under Al stress, some Al stress responding genes in Arabidopsis were used to test their expression patterns by qRT-PCR. The results can be put into four categories, according to the treatments with and without 50 μM AlCl_3_. Compared to those of wild type, three Al-upregulated genes of *AtMATE* (2.3-fold), *STOP1* (2.5-fold), and *PGIP2* (2.3-fold) were induced with relative expression abundances of over 2.3-fold in *GsMAS1* transgenic lines, whereas the RNA transcript of *WRKY46* sensitive to Al toxicity was less than half of that in the *GsMAS1* transgenic line. Under the treatment of 50 μM AlCl_3_, the RNA abundance of *AtMATE*, *AtALMT1*, *STOP1,* and *STOP2* in *GsMAS1* transgenic plants was much higher than that in wild type with over 27%, 82%, 19%, and 67% of relative expression, respectively (Figure 6). In addition, *AtALMT1* and *STOP2* under Al stress are significantly upregulated both in wild type and in *GsMAS1* transgenic plants with the largest absolute and relative expression (Figure 6). However, the RNA transcripts of *PGIP1* and *PGIP2* were less in *GsMAS1* transgenic lines than those in wild type of Arabidopsis at 50 μM AlCl_3_ treatment (Figure 6). Furthermore, *WRKY46* and *ALS3* sensitive to Al stress had less RNA transcripts at 50 μM AlCl_3_ treatment than those in the *GsMAS1* transgenic lines (Figure 6). All the results suggest that *GsMAS1* may enhance the tolerance to Al stress through the comprehensive effects of certain pathways in Arabidopsis.

## 3. Discussion

In this study, *GsMAS1* was cloned from the *Glycine soja* BW69 line. Sequence alignments indicated that putative GsMAS1 protein is highly homologous, with over 59% identity compared to those of AtAGL21, BRADI_5g12437v3, GmNMHC5, GmAGL15, GsMADS23L1 and MTR_2g009890 proteins (Figure 1a). The GsMAS1 protein possesses the typical structural features of a MADS-box protein with a conserved MEF2 MADS domain and the k-box domain, which was consistent with the structure of the Type II subfamily of MADS-box transcription factors in plants [79]. The GsMAS1 protein was localized to the nucleus of Arabidopsis protoplast cells (Figure 3). A similar result was found in GmNMHC5 protein, which shares 100% amino acids identity with GsMAS1 protein [90]. Therefore, we speculated that GsMAS1 protein may function as a MADS-box transcription factor and play certain roles in plants.

The phylogenetic analysis showed that GsMAS1 protein including GmNMHC5 protein belongs to the Type II subfamily of MADS-box members under the branch of the AGL17/AGL21 subfamily with a closer relationship to MADS TF proteins in soybean (Figure 1b) [90]. It has been reported that the members of AGL17 subfamily play significant roles in root system architecture construction and/or root development [94]. The *GsMAS1* gene, which is a constitutive expression gene with the richest transcripts in soybean root [90] (Figure 2), was quickly upregulated by AlCl_3_ treatment (Figure 2). Therefore, we speculate that *GsMAS1* is involved in the regulation of plant tolerance to Al stress.

To further investigate its function, *GsMAS1* was transformed into Arabidopsis to obtain the homologous lines to verify the Arabidopsis tolerance to Al stress. Compared with the control treatment, the elongation of main roots of wild-type and transgenic Arabidopsis plants was inhibited to a serious degree by aluminum stress along with the increase of AlCl_3_ concentration (Figure 5a). Under 50 μM AlCl_3_ treatment, the root lengths of transgenic Arabidopsis with a relative root elongation (RRE) of over 70% were larger than those of wild type with an RRE of 50%. Significant differences were also found between the AlCl_3_ treatments of 100 μM and 150 μM AlCl_3_ (Figure 5a). Similar resistant phenotypes to abiotic stress were also investigated in other TF genes in soybean. For example, *GmWRKY16* enhanced the tolerance of transgenic Arabidopsis plants to drought and salt stresses through an ABA-mediated pathway in *Arabidopsis thaliana* [95]. In this study, the contents of proline in *GsMAS1* transgenic lines and wild type were also determined to investigate the response of Arabidopsis plants to Al stress, and the results were consistent with those of previous reports (Figure 5c) [93,95]. The results suggest that *GsMAS1* transgenic Arabidopsis may promote plant tolerance to Al stress to a certain extent by the content changes of free proline accumulation.

In the present study, several Al-stress-response genes were used to investigate the molecular basis of *GsMAS1* tolerance to Al stress. Six Al-stress up-regulating genes and two Al-stress down-regulating genes indicated that they have different responses to Al stress with different RNA abundance, both in *GsMAS1* transgenic plants and wild type, under Al stress. On the basis of expression patterns, we speculate that *STOP1, AtALMT1,* and *STOP2* are the key genes, which may play a critical role in *GsMAS1* regulation pathways (Figure 6). The qRT-PCR analysis showed that *STOP1* was activated by *GsMAS1* and Al stress (Figure 6), which plays essential roles in proton and aluminum toxicities by regulating multiple genes in Arabidopsis [25,26,27,29,30]. The *AtALMT1* gene, which is the target gene regulated by *STOP1* and the *STOP2* gene, which is a homolog of *STOP1,* were significantly induced by *GsMAS1* expression in transgenic Arabidopsis plants with much higher transcription levels (Figure 6) [19,20,21,22,29]. Previous reports indicated that *STOP2* confers Al and low pH tolerance by activating transcription of several genes regulated by *STOP1* in Arabidopsis [29]. *AtMATE* and *AtALMT1* in Al stress response identified through work on STOP1 were upregulated by *GsMAS1* overexpression under the treatment of AlCl_3_ (Figure 6), which conferred aluminum tolerance in Arabidopsis independently by accumulation of aluminum-activated citrate and malate [19,20,21,22]. Moreover, *PGIP1* and *PGIP2*, which were downregulated by *STOP2* and associated with cell wall stabilization at low pH [29], were activated being expressed in *GsMAS1* transgenic lines under aluminum stress (Figure 6). As a negative regulator of *ALMT1*, *WRKY46* regulated aluminum-induced malate secretion in Arabidopsis [23], whereas *ALS3* protected the growing root from Al toxicity by redistributing accumulated Al away from sensitive tissues [96]. The RNA transcripts of *WRKY46* and *ALS3* both in wild type and *GsMAS1* transgenic lines under aluminum stress were less than those in the treatment of the control (Figure 6). The results indicated that *GsMAS1* enhanced resistance to Al toxicity by certain pathways in Arabidopsis. In addition, three genes, *AtMATE*, *STOP1,* and *PGIP2*, were upregulated by *GsMAS1* in Arabidopsis (Figure 6), which suggested that *GsMAS1* might participate in the regulation of other potential functions of the MADS genes in Arabidopsis.

Furthermore, phylogenetic analysis showed that GsMAS1 protein holds the same sequence as GmNMHC5 protein (Figure 1b). However, the non-coding sequences of *GsMAS1* are different from those of *GmNMHC5* (Additional file 1, Appendix A). The *GmNMHC5* gene induced by sucrose with a temporal and spatial expression pattern was rich in roots, nodules, and pods. Overexpression of *GmNMHC5* significantly accelerated lateral root development and nodulation in soybean [90]. Moreover, rich in roots and pods, *GsMAS1* enhanced resistance to Al stress in transgenic Arabidopsis without phenotype of root development (Figure 5a). The phenotype differences between soybean and Arabidopsis may be due to the differential activations and/or divergent pathways regulated by GsMAS1 and GmNMHC5 proteins, respectively. Therefore, further studies could focus on the potential roles in the regulation of *GsMAS1* promoter and the target genes interacting with *GsMAS1*.

The MADS family is one of the biggest transcription factor superfamilies and plays a fundamental role in almost every developmental process in soybean. Shu et al. (2013) found that there were 106 putative MADS-box genes by genome-wide survey and expression analysis, 72 genes among them being type II MADS-box TFs [72]. Fan et al. (2013) obtained 163 MADS genes from soybean genome by genome-wide expression analysis, and 115 MADS-box genes among them play potential function in seed development [36]. In recent years, researches on the MADS-box genes have focused on the regulations of plant growth and development, and flowering in soybean [90]. However, there are few reports for soybean MADS-box transcription factors/regulators related to abiotic and heavy metal stresses. More and more MADS-box transcription factors found in succession by the means of bioinformatics and genomics-wide analysis will provide the database to make profound studies for their roles of MADS-box TFs in the response to Al stress in soybean. Consequently, further study on the function of the *GsMAS1* gene will enrich our understanding of the mechanism of acid-tolerant aluminum in plants.

## 4. Materials and Methods

### 4.1. Plant Materials and Growth Conditions

The BW69 line of *Glycine soja* and Columbia-0 (Col-0) ecotype of Arabidopsis were both propagated and preserved by the Guangdong Subcenter of the National Center for Soybean Improvement (Guangzhou, China). The seeds of the BW69 line were germinated in vermiculite with room temperature set at 28/26 °C and the light time set as 14 h-light/10 h-dark under a light intensity of 110 μmol/(m^2^·s) [89,92]. The Arabidopsis seeds of Col-0 ecotype were germinated in nutrient soil (substrate: vermiculite = 3:1) with room temperature set as 24/22 °C and the light time set as 16 h-light/8 h-dark under a light intensity of 120 μmol/(m^2^·s) [89,92].

### 4.2. Bioinformatics Analysis of GsMAS1 Gene

The sequencing data from the gene expression profile resistant to acidic aluminum of the *Glycine soja* BW69 line were used to find the potential sequence of the *GsMAS1* gene. Sequence Blast was carried out in the NCBI (http://www.ncbi.nlm.nih.gov/) and Phytozome (http://phytozome.net/) databases to obtain the putative sequence of *GsMAS1* for gene cloning, predicted domains of GsMAS1 protein and the sequences of amino acids for the related homologous proteins. The software package DNAMAN was used to perform homologous sequence alignment, whereas MEGA (V6.0) software (Tokyo Metropolitan University, Tokyo, Japan) was used to analyze phylogenetic relations among all the MADS-box transcription factors [92].

### 4.3. Analysis of GsMAS1 Expression Patterns

To analyze tissue expression pattern of *GsMAS1*, the seeds of the BW69 line with cut-coat on the back of hilum were sowed in the farm field of the campus at South China Agricultural University. When the wild soybean plants began to pod, the samples of flower and young pod were taken at the R_2_ stage [89,92], and then frozen in liquid nitrogen and stored at −80 °C in an ultra-low temperature refrigerator [92]. The samples of root, stem, and leaf were taken from the young seedlings of the BW69 line [89].

To investigate the response of *GsMAS1* to acid aluminum stress, the cut-coat seeds of the BW69 line were prepared and sterilized with 3% hydrogen peroxide for 30 s, rinsed 3–4 times with distilled water and sprouted in autoclaved vermiculite. When the leaf blades of soybean seedlings had completely unfolded, the seedling vermiculite on the surface of leaves was rinsed with ultrapure water. The cleaned seedlings pre-cultured in nutrient solution (pH 5.8) for 48 h were then transferred to a simple calcium solution of 0.5 mM CaCl_2_ (pH 4.3) for pretreatment after 24 h. The concentrations of AlCl_3_ (0.5 mM CaCl_2_, pH 4.3) solution were set as 0 μM, 15 μM, 30 μM, 50 μM, 75 μM, and 100 μM individually [97]. Eighteen seedlings were treated at each concentration for three replicates. Samples of 2 mm-long root tips were harvested after AlCl_3_ treatment for 6 h, wrapped in foil, frozen in liquid nitrogen, and then stored at −80 °C in an ultra-low temperature refrigerator.

To detect the temporal expression pattern of *GsMAS1*, the seedlings of the BW69 line were prepared in 0.5 mM CaCl_2_ solution (pH 4.3) in the same way above and separately sampled at 0 h, 6 h, and 12 h. During the treatment process of 50 μM AlCl_3_ (0.5 mM CaCl_2_, pH 4.3) for 48 h, the samples of 2 mm-long root tips from young seedlings were taken at 2 h, 4 h, 6 h, 8 h, 12 h, 24 h, and 48 h, individually [97], and then frozen in liquid nitrogen and stored at −80°C in an ultra-low temperature refrigerator.

### 4.4. Real-time Quantitative PCR

TransZol Up (Trangen Biotech, Beijing, China) was used to extract total RNA from different samples. First-strand cDNA was synthesized using EasyScript One-Step gDNA Removal and cDNA Synthesis SuperMix Kit using 1.5 μg RNA as a template. Quantitative real-time PCR was performed on a CFX96 Real-Time PCR Detection System device (Bio-Rad, Hercules, CA, USA) using the SsoFast EvaGreen Supermix Kit (Bio-Rad, Shanghai, China). All the reactions were carried out in 20 μL volumes containing 1 μL cDNA as a template. Primers are listed in Appendix A (Additional file 3) for ACT3-F and ACT3-R, qGsMAS1-F and qGsMAS1-R, and At-TUB-F and At-TUB-R. The following procedure was used for qRT-PCR: 94 °C for 3 min; 40 cycles of denaturation at 94 °C for 10 s, annealing at 59 °C for 10 s, elongation at 72 °C for 30 s. Data were analyzed using the 2^−ΔΔCT^ method as described above [98].

### 4.5. Subcellular Localization Analysis of GsMAS1 Protein

To investigate the subcellular localization of GsMAS1 protein, the full-length cDNA without the stop codon of GsMAS1 was inserted into the restriction sites of *Kpn*I and *Bam*HI to construct a pYL322-d1-eGFP-GsMAS1 vector using the specific primers of C-GsMAS1-F and C-GsMAS1-R (Additional file 3, Appendix A) [92]. The CDS (coding sequence) sequence of *GsMAS1* was inserted into the downstream of the GFP sequence by recombinant DNA technology. The vectors pYL322-d1-eGFP and pYL322-d1-eGFP-GsMAS1 were then transformed into Arabidopsis protoplasts by a PEG-mediated method and cultured in the dark for two days [21]. The expression signal of GFP was observed and photographed under a confocal laser scanning microscope (Carl Zeiss, Oberkochen, Germany).

### 4.6. Cloning of GsMAS1 Gene

The gene of Gene ID: 100,805,092 was selected from the gene expression profile resistant to aluminum stress of the *Glycine soja* BW69 line, and then the NCBI database was used to search for the sequence information for the candidate gene. According to the obtained sequence from the database of NCBI, the CDS sequences of the *GsMAS1* gene were amplified from cDNA of *Glycine soja* BW69 roots with specific primers GsMAS1-F and GsMAS1-R (Additional file 3, Appendix A). Total RNA extraction, cDNA generation and RT-PCR amplification were carried out according to the methods described in detail previously [89]. The PCR product was purified by 1% agarose gel electrophoresis (GenStar Kit, Genstar Development Company, Calgary, AB, Canada) and then purified by Tiangen Rapid DNA Ligation Kit (Beijing, China). The purified fragment for *GsMAS1* was inserted into the multi-cloning site of pLB vector to construct pLB-GsMAS1 vector with the reference of pLB zero background rapid cloning kit (TIANGEN, Beijing, China). Clones of *E. coli* transformed with the pLB-GsMAS1 vector using the heat-shock method were identified by PCR and enzyme digestion by methods previously described in detail [89]. The positive clones were sequenced to obtain the full cDNA sequence of *GsMAS1* (Sangon, Shanghai, China).

### 4.7. Construction of Plant Expression Vector

According to the sequence information, the ORF sequence of 723 bp length of the *GsMAS1* gene was inserted into the *Xba*I and *Sac*I sites of plant-expression vector pTF101.1-GFP by homologous recombination technology to construct the fusion expression vector of pTF101.1-GsMAS1. The expression of the *GsMAS1* gene was driven by the 35 S promoter in *E. coli* cells. The experimental operations PCR amplification, DNA segregation and purification, and fusion vector transformation for the target fragment were carried out by the methods for *GsMAS1* cloning described above in detail. All the monoclonal clones were identified by PCR and restriction enzyme digestion. The positive clones were sent to a biotechnology company (Sangon, Shanghai, China) for sequencing, and the sequence result was then confirmed using the NCBI database [91].

### 4.8. Heterologous Expression of GsMAS1 in Arabidopsis

The ORF of *GsMAS1* was inserted into the *Xba*I and *Sac*I sites to form a pTF101.1-GsMAS1 vector using the primers p-GsMAS1-F and p-GsMAS1-R (Additional file 3, Appendix A) by the homologous recombination method. The recombinant plasmids were transformed into the competent cells of *Agrobacterium tumefaciens* GV3101 by the electroporation method. Arabidopsis genetic transformation was carried out using the floral dip method to obtain transgenic plants [99]. After the siliques matured, seeds of T_0_ generation were harvested, dried and then vernalized at 4 °C for further identification and generation. When two pieces of true leaves are fully unfolded, the T_1_ Arabidopsis seedlings were initially screened with 10 mg/L glufosinate and then further identified at the DNA and RNA levels [89]. After the positive seedlings of transgenic Arabidopsis plants matured, the seeds were individually harvested to form transgenic lines. Homozygous transgenic lines of T_3_ generation by herbicide application of 10 mg/L glufosinate and PCR identification were produced to investigate the *GsMAS1*phenotypes tolerant to Al stress [89].

### 4.9. Phenotypic Identification of GsMAS1 Transgenic Lines

In the present study, 13 positive transgenic plants obtained in succession by herbicide application and PCR identification were propagated to gain the *GsMAS1* transgenic lines of T_3_ generation. To obtain homozygous lines, the Arabidopsis seedlings in T_3_ generation of 13 *GsMAS1* transgenic lines underwent molecular identification. The specific band of PCR products detected by agarose gel electrophoresis indicated that the ORF sequence of *GsMAS1* was successfully inserted into the Arabidopsis genome. The Arabidopsis seeds of wild-type and *GsMAS1*transgenic lines of T_3_ generation sterilized with 70% ethanol for 2 min, 1% sodium hypochlorite for 5 min, and sterilized water three times were spotted on 1/2 MS solid medium (pH 5.8) in one-time culture dish sealed with the microporous air-permeable sealer. After incubation at 4 °C for 2–3 days, Arabidopsis seeds in all the culture dishes were placed in a vertical light incubator with the conditions of light time 16 h/8 h (day/night), temperature 22–24 °C, humidity 50%. When the root length of Arabidopsis plants was up to about 1 cm, the seedlings with almost the same root length were selected and transferred to 1/2 MS solid agar medium containing different AlCl_3_ concentrations set as 0 μM, 50 μM, 100 μM, 150 μM, and 200 μM (0.5 mM CaCl_2_, pH4.5). After being cultured for 7 days, all the treatments for phenotypes tolerant to Al stress were photographed with the root length being measured by Image J software [100]. The index of relative elongation of main root (%) was used to evaluate root changes of Arabidopsis seedlings under Al stress [100]. The seedlings under the treatments of 0 μM and 50 μM AlCl_3_ of *GsMAS1* transgenic lines and wild-type plants were used to determine contents of free proline by the methods described in detail previously [97].

### 4.10. Statistical Analysis

All the experiments for the analysis of expression pattern, molecular identification, phenotype tolerant to Al stress and regulation pathway of *GsMAS1* were carried out with three independent biological replicates. All the data were presented as mean ± SE. An LSD test (*p* = 0.05 or *p* = 0.01) was carried out to investigate the differences between observation values [89].

## 5. Conclusions

We investigated a soybean *GsMAS1* gene encoding a MADS-box transcription factor from BW69 line of *Glycine soja*. The *GsMAS1* gene was induced by Al stress and rich in roots with a constitutive expression pattern in soybean. The GsMAS1 protein was located in the nucleus. Ectopic overexpression of *GsMAS1* in Arabidopsis enhanced the tolerance of transgenic plants to Al stress with free proline accumulation. The molecular investigation indicated that the enhanced tolerance to Al toxicity of *GsMAS1* transgenic Arabidopsis was caused by the comprehensive role played by some critical genes in responding to Al stress. The results suggested that GsMAS1 protein may increase resistance to Al toxicity through certain pathways related to Al stress in Arabidopsis and provide a scientific basis for soybean molecular breeding.

## Figures and Tables

**Figure 1 ijms-21-02004-f001:**
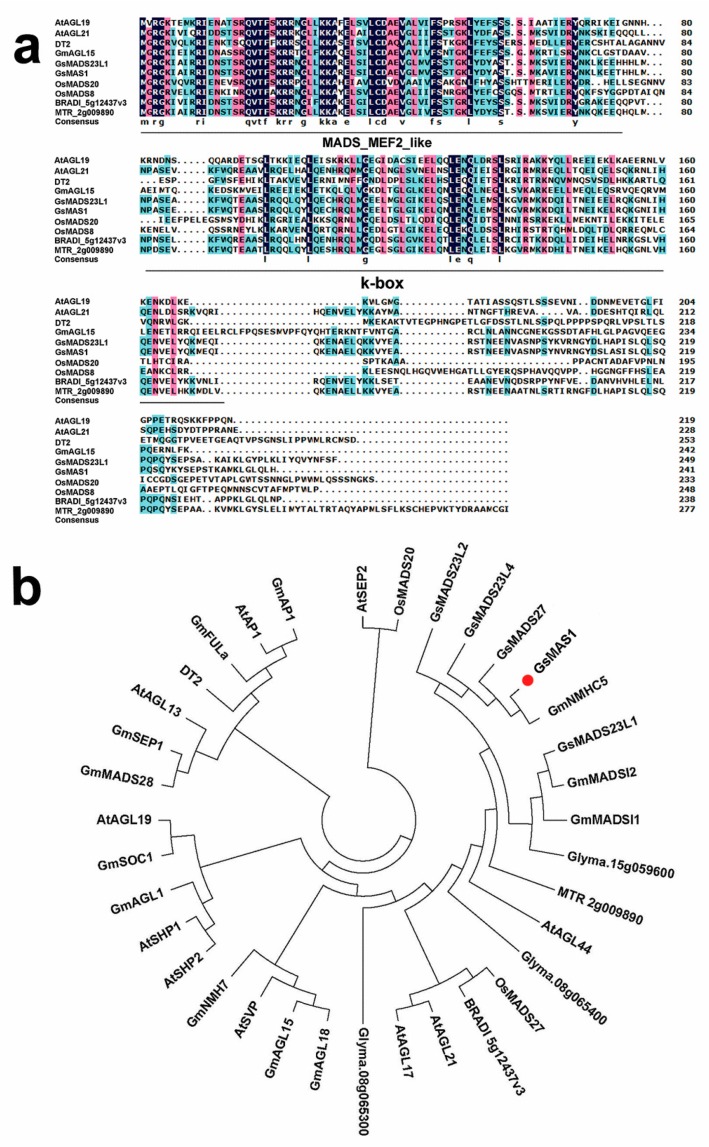
Homology analysis of GsMAS1 protein and other MADS-box transcription factors in plants. (**a**) Sequence alignment of GsMAS1 protein and MADS-box transcription factors. (**b**) Phylogenetic analysis of GsMAS1 and MADS-box transcription factors. The sequence alignment of amino acids was carried out by using the software DNAMAN6.0. The phylogenetic tree was constructed by the software MEGA (V6.0) with the neighbor-joining method. The amino acid sequences of MADS-box transcription factors are from the databases of the National Center for Biotechnology Information (NCBI) (https://www.ncbi.nlm.nih.gov/) and Phytozome (http://phytozome.net/). The information of accession number and species for MADS-box transcription factors is as follows. The proteins of AtAGL13 (AAC49081), AtAGL17 (OAP11731), AtAGL19 (AAG37901), AtAGL21 (NP_195507.1), AtAGL44 (NP_179033.1), AtAP1 (CAA78909), AtSHP1 (OAP06129), AtSEP2 (AAF61626), AtSHP2 (NP_850377), and AtSVP (OAP09056) are from *Arabidopsis thaliana*. The proteins of Dt2 (NP_001340272), Glyma.08g065300 (KRH42047), Glyma.08g065400 (XP_014634038), Glyma.15g059600 (KRH10635), GmAGL1 (NP_001304521), GmAGL15 (NP_001237033), GmAGL18 (XP_006575259), GmAP1 (XP_003547792), GmFULa (ahi43155), GmMADS28 (NP_001236390), GmMADSI1 (XP_014623536), GmMADSI2 (XP_025981482), GmNMH7 (NP_001236857), GmNMHC5 (XP_006593452), GmSEP1 (AAZ86071) and GmSOC1 (NP_001236377) are from *Glycine max*, whereas GsMADS23L1 (XP_028204324), GsMADS23L2 (XP_028187089), GsMADS23L4 (XP_028187090), GsMADS27 (RZB82838), and GsMAS1 are from wild soybean (*Glycine soja*). The proteins of OsMADS8 (Q9SAR1), OsMADS20 (Q2QQA3) and OsMADS27 (XP_015626695) are from *Oryza sativa*; MTR_2g009890 (AES63546) is from *Medicago truncatula*, BRADI_5g12437v3 (KQJ82998) is from *Brachypodium distachyon*.

**Figure 2 ijms-21-02004-f002:**
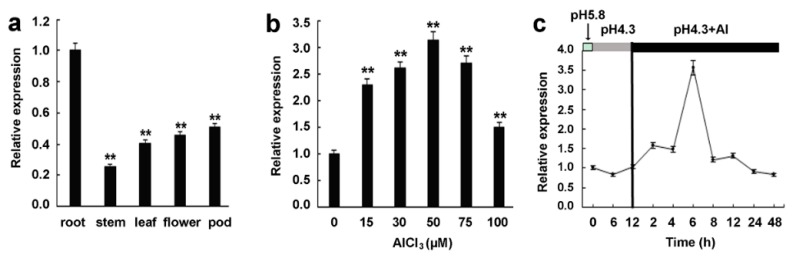
Expression patterns of *GsMAS1* in tissues and under acidic aluminum exposure. (**a**) Expression pattern analysis of *GsMAS1* in soybean tissues. Samples of roots, stems, and leaves are from young seedlings; flowers and pods were taken at the period of blooming and 15 days after soybean flowering, respectively. (**b**) Expression pattern of *GsMAS1* under the AlCl_3_ treatments. After seed germination for two days, the seedlings of wild soybean (BW69 line) were transferred to AlCl_3_ solutions, which were set as 0, 15, 30, 50, 75, and 100 µM (pH 4.3, 0.5 mM CaCl_2_). After AlCl_3_ treatments for 6 h, the 6-cm-long roots of seedlings were harvested to analyze the expression levels of *GsMAS1*. (**c**) Temporal expression pattern of *GsMAS1* under acidic aluminum exposure. The 2-day seedlings after germination were cultured in a solution of 0.5 mM CaCl_2_ (pH 4.3) for 24 h, and then they underwent the 50 µM AlCl_3_ treatment (pH 4.3, 0.5 mM CaCl_2_). The 6-cm-long roots were taken from the seedlings at the processing time nodes of 2, 4, 6, 8, 12, and 24 h. The qRT-PCR was carried out to assess the transcript abundance of *GsMAS1* by the 2^−∆∆Ct^ method with *ACT3* as an internal control [92]. Three independent biological experiments were carried out to calculate the relative expression value of *GsMAS1*. Data are means ± SE, and the asterisks (**) represent a significant difference (*p* = 0.01).

**Figure 3 ijms-21-02004-f003:**
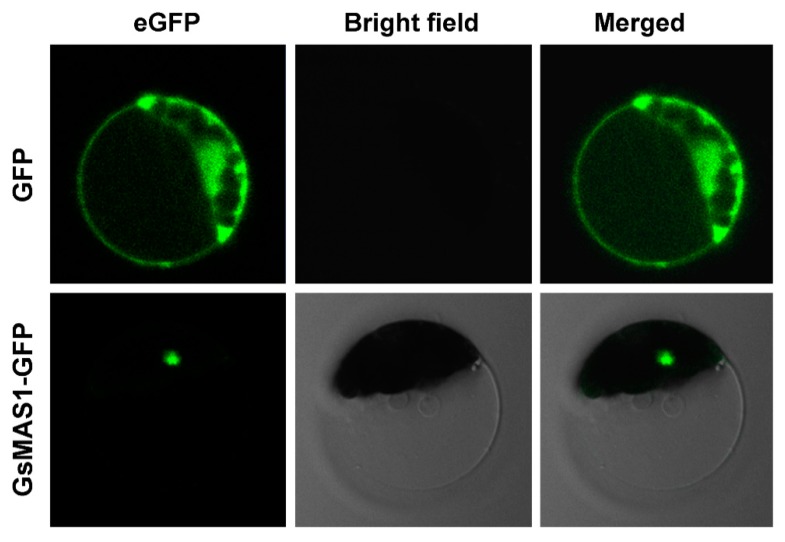
Subcellular localization of GsMAS1 protein.

**Figure 4 ijms-21-02004-f004:**
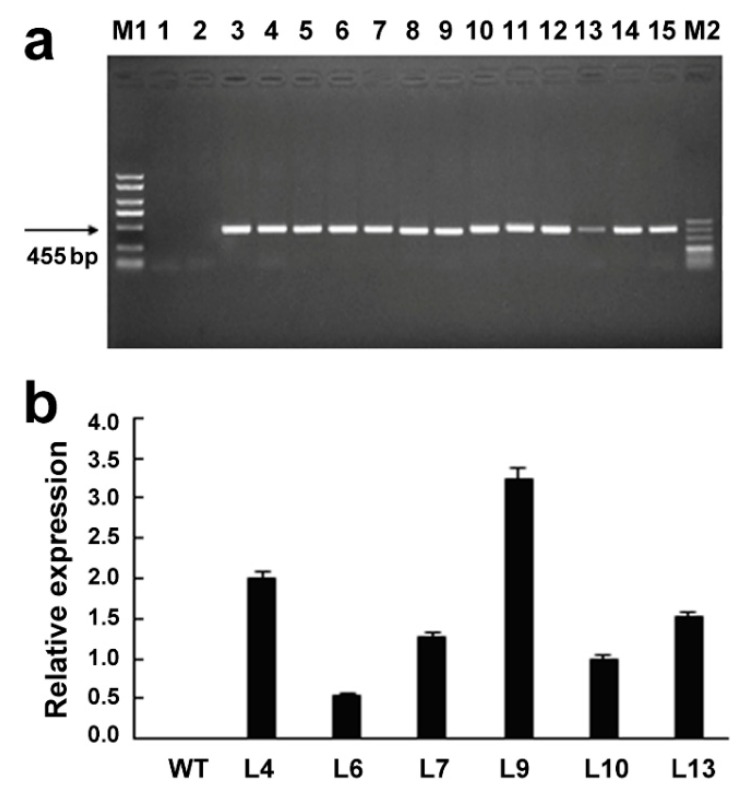
Molecular identification for *GsMAS1* transgenic lines in Arabidopsis. (**a**) PCR identification of *GsMAS1* transgenic plants. (**b**) qRT-PCR identification of *GsMAS1* transgenic lines. The 3-week-old seedlings of Arabidopsis were used to identify *GsMAS1* transgenic plant and lines. M1: DNA maker DL5000; M2: DNA marker DL500; Lanes 1–15: PCR products with different DNA templates set as ddH_2_O for lane 1, genomic DNA from wide type for lane 2, genomic DNA from *GsMAS1* transgenic plants for lanes 3–15. WT: wild type of Arabidopsis Columbia-0; L4 to L13: six transgenic lines of *GsMAS1* in T_3_ generation. Data are means ± SE. Error bars represent the standard error of four replicates.

**Figure 5 ijms-21-02004-f005:**
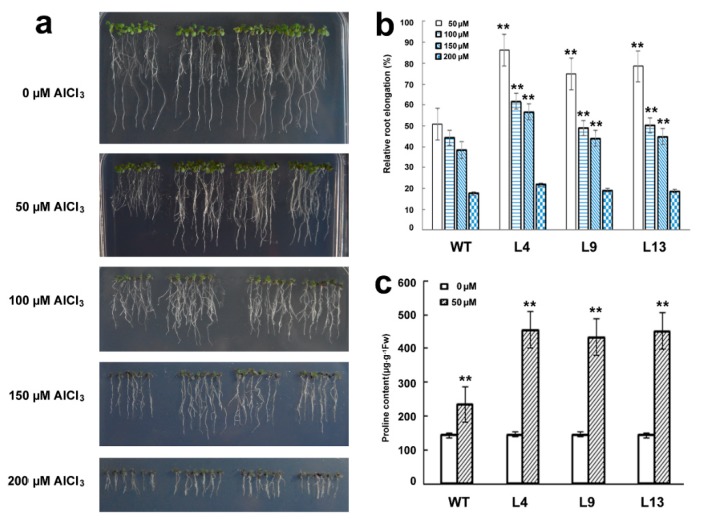
*GsMAS1* enhanced the resistance of Arabidopsis plants to Al stress (**a**) The phenotypes of *GsMAS1* transgenic lines tolerant to Al stress. (**b**) Statistical analysis of relative root elongation. (**c**) The determination of free proline content. WT: wild type of Arabidopsis (Col-0); L4, L9, L13: *GsMAS1* overexpression transgenic lines of T_3_ generations. The two-day seedlings after seed germination were transferred to culture medium containing AlCl_3_ (pH 4.3, 0.5 mM CaCl_2_) at 0, 50, 100, 150, and 200 µM, individually. After being cultured for seven days, phenotypic images and contents of free proline for the *GsMAS1* transgenic lines were recorded and/or determined for statistical analysis. Data were represented as the means ± SE of three biological replicates (Student’s test, ** *p* = 0.05). The programs of Image J, SPSS20, and EXCEL2000 were used to measure and/or analyze the data.

**Figure 6 ijms-21-02004-f006:**
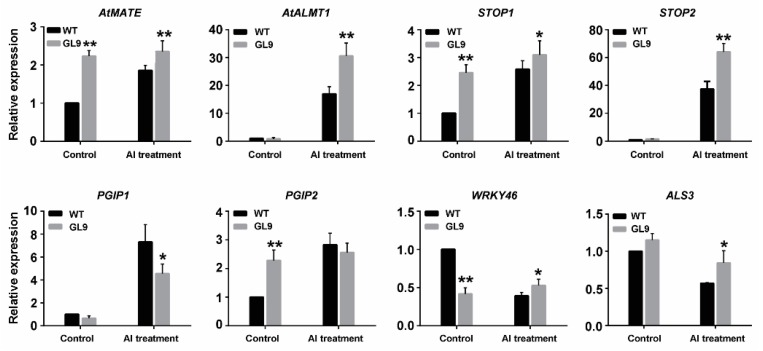
Expression patterns of Al stress responsive genes regulated by *GsMAS1.* Arabidopsis seeds of WT and the *GsMAS1* transgenic line L9 of T_3_ generation were sown in 1/2 MS (Murashige-Skoog) culture media and cultured in the chamber room. The two-week-old seedlings were subjected to the treatment of solutions with or without 50 µM AlCl_3_ (pH 4.3, 0.5 mM CaCl_2_) for 24 h. The seedling samples from Arabidopsis plants were taken to extract total RNA. The transcription abundance of investigated genes was quantified by qRT-PCR using *ACT3* as the inner reference gene. The quantitative variation between the examined replicates was determined by the 2^−∆∆Ct^ method. The details of specific primers for *GsMAS1*, *ACT3* and measured genes are listed in Appendix A (Additional file 3). WT: wild type; GL9: *GsMAS1* transgenic line L9 of T_3_ generation. The data were represented as the mean ± SE of three biological replicates (t-test, ** *p* = 0.05).

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
