# Peer review of "GsMAS1 Encoding a MADS-box Transcription Factor Enhances the Tolerance to Aluminum Stress in Arabidopsis thaliana"

_ijms, 2020, doi:10.3390/ijms21062004_

Round 1

Reviewer 1 Report

This manuscript described elucidating molecular mechanisms of GsMAS1 gene, which encodes one of MADS box transcription factors and is associated with Aluminum (AL) tolerance in soybean. qRT-PCR and phenotyping of transgenic lines revealed that high expression of the GsMAS1 strongly enhanced AL tolerance in Arabidopsis plants. The aim of this manuscript is important to understand molecular regulation of AL tolerance in soybean. However, experimental results and descriptions in this manuscript are not enough to draw your conclusions. The authors should revise several descriptions in this manuscript described below.

1) L.125-L.132. The authors should describe detailly the reason why you select the GsMAS1 gene. There were a lot of unpublished and not shown data in this section. You should indicate individual experimental results in this section as new figures or tables. I would like to know the reason why you focused on the GsMAS1 gene.

And, the authors should describe scientific evidences about strong AL resistance of BW69. You should indicate phenotypes of BW69 under AL treatment cultivation conditions as new figure. The authors could also cite your previous paper such as Ma et al. 2018. BMC Plant Biol. to explain it.

2) L.229-L.259. This section described methods to develop transgenic Arabidopsis plants. Therefore, the authors should move these texts to the Materials and Methods section. You could delete these sentences if these descriptions were duplicated with the Materials and Methods section.

And, L.260. After revising these descriptions, 'To investigate the transgenic lines' should be changed to 'To investigate the 13 transgenic lines'.

3) In Figure 5. The authors should evaluate total root weights of each plant, as well as their root length and proline contents.

4) In Figure 6. The authors investigated expression patterns of eight Al stress responsible genes. To investigate differentiation of whole genes, the authors should perform RNA-seq in this section. As you mentioned at the Discussion section, expression of ABA related genes also changed under several abiotic stress. And, auxin related genes and WRKY genes were associated with abiotic stress tolerances. You could discuss these points based on the results of RNA-seq.

5) L.371-L.372. The authors should indicate full names of two abbreviations of MDA and ROS.

Author Response

Dear Reviewer:

Thanks very much for the good suggestions and comments. The relative results were embellished in detail to support our conclusions in the revised manuscript.  The updae manuscript was revised point to point throughout the text as your comments.

  Qibin MA

Reviewer 2 Report

General comments:

The work of Zhang et al. is interesting and new but several points listed below require clarifications.

In the Results part, many details are given that should be located in the Materials and Methods.

I suggest that authors check the bibliographic references in the main text. There are 107 references (a lot for a research paper!) but there are confusions in many of them.

Specific comments:

Introduction :

The literature about aluminum tolerance in general and STOP1 in particular must be developed.

Line 67 : reference 29 is wrong.

Line 102 : How many MADS-box genes in soybean ?

Results:

Line 125-133 : It is not clear how the authors identified GsMAS1: the authors indicate that it comes from Glycine soja, but the NCBI reference (NM_001254560) corresponds to a Glycine max gene.

The method should better explained. Was the gene expression data from the authors? How the plants were grown for the data that the authors analysed ? Are the data accessible (the the NCBI reference NM_001254560 corresponds to GsMAS1, not to a transcriptomic database)? Unfortunately, the Materials and Methods does not help.

Line 131: the growth conditions are from the authors or from the “unpublished data?

Line 127: The authors give the NCBI reference NM_001254560 as the identity of their gene identified in Glycine soja. This genes is published as « A Novel Sucrose-Regulatory MADS-Box Transcription Factor GmNMHC5 Promotes Root Development and Nodulation in Soybean (Glycine max [L.] Merr. ». 

Chapters 2.3, 2.4, 2.5, 2.6 are full of details that should be in the Methods section. 

Chapter 2.5: Indicate which promoter drives the expression of GsMAS1 (pTF101.1-GsMAS1 vector). There is nothing about it in the Materials and Methods. 

Figure 4 and 5: Are the lines homozygous for the GsMAS1 transgene? If yes, how were they selected as homozygous? 

Chapter 2.8: Explain the rational of measuring free proline content; what is the link between free proline and tolerance to aluminum?

Lines 312-314: “…free proline promoted the tolerance to Al stress of GsMAS1 transgenic Arabidopsis…”. The authors observe a positive correlation between free proline content and tolerance to aluminum of GsMAS1 transgenic lines, but how do they know that it is a causal effect? 

Figure 5C: Why the three transgenic lines have the same free proline content although they do not express GsMAS1 at the same level? 

Figure 5: Does the better root growth of the transgenic lines under aluminum correlate with malate and citrate exudation?

Does a KO mutant of the closest Arabidopsis homologues of GsMAS1 display a particular root phenotype under aluminium condition?

Chapter 2.9: I suggest the authors to compare the transgenic lines with the WT, not the transgenic in 0 versus +Al.

Figure 6: It is necessary to perform a statistical analysis of these results to know which are significant.  

Discussion:

Lines 347-348: What is the meaning of the following sentence: “The biochemical characteristic analysis indicated GsMAS1 protein held the feature of no transcriptional activation in yeast cells (data not shown)”? Either the authors show the corresponding experiment, either they do not mention it.

Lines 348-349: The authors should mention that Liu et al. (Int. J. Mol. Sci. 2015, 16, 20657-20673) already shown the nuclear localisation of GmNMHC5 (from Glycine max) which shares 100% amino acids identity with GsMAS1.

Lines 352-354: The authors should mention a similar result published by Liu et al. (Int. J. Mol. Sci. 2015, 16, 20657-20673). 

Lines 355-357: Compare with results obtained by Liu et al. (Int. J. Mol. Sci. 2015, 16, 20657-20673). 

Lines 357-360: I don’t understand the logic of these assertions.  

Line 373: The two publications mentioned (N°91 & 92) are not appropriate. 

Lines 379-395: I think that before asserting that GsMAS1 confers tolerance to aluminum through the up-regulation of different genes/pathways, the authors should first validate the results of Figure 6 by independent experiments and by a statistical analysis.  

Line 471: This chapter is wrongly entitled “Biochemical characteristics analysis of GsMAS1 protein” because there is no biochemical characterization of GsMAS1 in this work.

Author Response

Dear Reviewer:

Thanks very much for the good suggestions and comments. The points listed below were clarified one by one in the text. Many details were moved to the corresponding positions in the Materials and Methods. All the references were checked and formatted to meet the requirement of IJMS.

  Qibin MA

Round 2

Reviewer 1 Report

The authors clearly revised several descriptions in this manuscript according to my previous comments. However, the authors didn’t revise the remain points about requiring additional experiments such as investigation of root weight in Fig 5 and whole transcriptome analysis in Fig 6. If you cannot perform or don’t need to perform these experiments, could you please explain those reasons?

Author Response

Dear reviewer,

Thank you very much for your great suggestions and comments. I agree with you that the required experiments will provide us the effective results which will perfect our manuscript. However, we can’t carry out the experiments for some objective conditions by the epidemic situation of the novel coronavirus COVID-19 in China as follows: 1) All students are not allowed to go back to campus by government until the COVID-19 epidemic situation is subsided; 2) All laboratories are not allowed to carry out relevant research work temporarily; 3) The whole transcriptome analysis can only be completed by biotech companies which are still not back to work for the time being.

We can't judge when the epidemic of coronavirus COVID-19 in China will be over, when students can return to campus and biological companies can begin to work. Therefore, I am very sorry that we can't provide relevant data to support our manuscript. If you have any other questions, please feel free to contact me.

Best regards,

Qibin MA

Reviewer 2 Report

The authors have done some improvement of their manuscript. However there are still some points that need to be revised.

Introduction:

Several major bibliographic references about the role of STOP1 and ALMT1 in Al tolerance are missing.

Results:

The following sentences should be in the Materials and Methods section :

Lines 138-139 : « the product of RT-PCR was ligated to pLB Vector (TIANGEN, Beijing, China). »

Lines 140 : (TaKaRa, Dalian, China) ;

Lines 241-246 : « In the present study, 13 positive transgenic plants obtained in succession by herbicide application and PCR identification were propagated to gain the GsMAS1 transgenic lines of T3 generation. To obtain homozygous lines, the Arabidopsis seedlings in T3 generation of 13 GsMAS1 transgenic lines underwent molecular identification. The specific band of PCR products detected by agarose gel electrophoresis indicated that the ORF sequence of GsMAS1 was successfully inserted into the Arabidopsis genome (Figure 4a). »

Figure 2a: Is the expression of GsMAS1 in roots not significant?

Line 292: In their answer the authors argue “Previous reports showed that proline accumulation can enhance maize tolerance to aluminum stress by minimizing accumulation of lipid peroxides.” This result/reference could be a good introduction to Chapter 2.6.

Figure 6: For the statistical analysis, I suggest again the authors to compare the transgenic lines with the WT grown in the same condition.

Line 525: it is written “All the data were presented as mean ± SD”. But in figures 2, 4, 5 and 6 it is the mean ± SE.

Discussion

Lines 343-344: The sentence “In a previous study, GmbHLH30, with a constitutive expression pattern, was rich in root and also up-regulated under Al stress [76].” should be removed; otherwise the following one makes a strange assumption with it.

Materials and Methods:

Line 428: replace “umbilical cord” by the appropriate botanical term.

Line 478: Replace “polyclonal sites” by “multi-cloning site”.

Line 488: I guess the sentence should be: “The expression of the GsMAS1 gene was driven by the 35S promoter.”

Chapter 4.5: Why removing the stop codon of GsMAS1 if it is fused to GFP at its C-terminal end?

Line 499: The reference for the “floral dip method” is wrong; this is : Plant J 16: 735-743 (1998)

Line 500: In arabidopsis the pods are named siliques.

Line 509: Did the authors have really sterilized their seeds with “mercury chloride”???

Line 517: The reference for the “Image J software” is wrong.

Author Response

Dear reviewer,

Thank you very much for your comments. The figure 6 was update in the revised version. All the references were checked and renewed thorough the text. The questions and comments had been explained point by point and modified in detail in the “Responses” and the revised manuscript, respectively. If you have any questions regarding this submission, please feel free to contact me.

Best regards,

Qibin MA

Round 3

Reviewer 1 Report

Thank you for sending your responses to my previous comments.
I understand your difficult situations to carry out additional experiments.
I hope everything is well with you.

Author Response

Dear reviewer,

Thank you very much for your understanding. We will improve the manuscript as soon as possible to meet the requirement of the journal.

Best regards,

Qibin MA